# Expanded Performance Comparison of the Oncuria 10-Plex Bladder Cancer Urine Assay Using Three Different Luminex xMAP Instruments

**DOI:** 10.3390/diagnostics15141749

**Published:** 2025-07-10

**Authors:** Sunao Tanaka, Takuto Shimizu, Ian Pagano, Wayne Hogrefe, Sherry Dunbar, Charles J. Rosser, Hideki Furuya

**Affiliations:** 1Samuel Oschin Comprehensive Cancer Institute, Cedars-Sinai Medical Center, Los Angeles, CA 90048, USA; sunao.tanaka@cshs.org (S.T.); takutea19@gmail.com (T.S.); charles.rosser@cshs.org (C.J.R.); 2Population Sciences in the Pacific Program, University of Hawaii Cancer Center, Honolulu, HI 96813, USA; pagano@hawaii.edu; 3Nonagen Bioscience Corp., Los Angeles, CA 90010, USA; wrhogrefe@gmail.com; 4Luminex Corp., Austin, TX 78727, USA; sherry.dunbar@diasorin.com; 5Department of Urology, Cedars-Sinai Medical Center, Los Angeles, CA 90048, USA; 6Department of Biomedical Sciences, Cedars-Sinai Medical Center, Los Angeles, CA 90048, USA

**Keywords:** bladder cancer, fluorescence, multiplex immunoassay, magnetic bead, performance, dynamic range, xMAP technology

## Abstract

**Background/Objectives**: The clinically validated multiplex Oncuria bladder cancer (BC) assay quickly and noninvasively identifies disease risk and tracks treatment success by simultaneously profiling 10 protein biomarkers in voided urine samples. Oncuria uses paramagnetic bead-based fluorescence multiplex technology (xMAP^®^; Luminex, Austin, TX, USA) to simultaneously measure 10 protein analytes in urine [angiogenin, apolipoprotein E, carbonic anhydrase IX (CA9), interleukin-8, matrix metalloproteinase-9 and -10, alpha-1 anti-trypsin, plasminogen activator inhibitor-1, syndecan-1, and vascular endothelial growth factor]. **Methods**: In a pilot study (N = 36 subjects; 18 with BC), Oncuria performed essentially identically across three different common analyzers (the laser/flow-based FlexMap 3D and 200 systems, and the LED/image-based MagPix system; Luminex). The current study compared Oncuria performance across instrumentation platforms using a larger study population (N = 181 subjects; 51 with BC). **Results**: All three analyzers assessed all 10 analytes in identical samples with excellent concordance. The percent coefficient of variation (%CV) in protein concentrations across systems was ≤2.3% for 9/10 analytes, with only CA9 having %CVs > 2.3%. In pairwise correlation plot comparisons between instruments for all 10 biomarkers, R^2^ values were 0.999 for 15/30 comparisons and R^2^ ≥ 0.995 for 27/30 comparisons; CA9 showed the greatest variability (R^2^ = 0.948–0.970). Standard curve slopes were statistically indistinguishable for all 10 biomarkers across analyzers. **Conclusions**: The Oncuria BC assay generates comprehensive urinary protein signatures useful for assisting BC diagnosis, predicting treatment response, and tracking disease progression and recurrence. The equivalent performance of the multiplex BC assay using three popular analyzers rationalizes test adoption by CLIA (Clinical Laboratory Improvement Amendments) clinical and research laboratories.

## 1. Introduction

In 2022, bladder cancer (BC) was newly diagnosed in 614,000 individuals worldwide and was associated with 220,000 deaths, with a 4-fold higher cumulative risk for males than females, and a higher incidence in European and North American populations than other regions [1,2]. Approximately 75% of the 83,000 BC cases diagnosed annually in the USA are non-muscle-invasive disease (NMIBC) that require years of ongoing surveillance for disease recurrence/progression after initial bladder resection with or without Bacillus Calmette–Guerin (BCG) therapy [3,4]. Muscle-invasive BC requires more aggressive initial treatment and long-term follow-up. Cystoscopy and voided urine cytology remain the gold standards for evaluating BC status [3,4]. Cystoscopy is invasive, expensive, and has significant associated risks (e.g., infection, trauma). Performing cytology on voided urine samples is noninvasive, comparatively inexpensive, and has high specificity for BC but also limited sensitivity, especially with low-grade and early-stage disease [5]. Analyzing protein biomarker profiles in urine samples has evolved into a noninvasive means to accurately identify BC, categorize disease risk, and evaluate treatment success [6].

There are no screening programs in common use for detecting BC, and diagnosis often follows patient presentation for gross (macroscopic) hematuria (i.e., cystoscopy, cytology, and upper tract imaging) or BC surveillance (i.e., cystoscopy and cytology). BC is a molecularly diverse disease, and no individual urine biomarker has yet been identified that can reliably identify and track disease, or predict the likelihood of recurrence or treatment-responsiveness to BCG [6,7,8,9]. Furthermore, urinary concentrations of some protein-based BC markers (e.g., nuclear matrix protein 22 [10] and bladder tumor antigen [11]) are increased in non-cancerous scenarios such as urinary tract inflammation, which can erroneously cause positive cancer determinations.

Multiplex assays that simultaneously evaluate diverse BC biomarkers in urine specimens increase test accuracy for identifying cancers with varied etiology and presentation, predicting treatment response, and accurately tracking therapy effectiveness [5]. This approach can quickly and noninvasively create detailed patient-specific BC protein signatures that can assist BC diagnosis, disease characterization and staging, and personalized treatment planning with the goal of improving patient outcomes [5]. Oncuria^®^ (Nonagen Bioscience Corp., Los Angeles, CA, USA) is a bead-based multiplex fluorescence immunoassay that simultaneously measures the concentrations of 10 protein biomarkers in urine [12,13,14,15]. Analyte levels are mathematically integrated with patient characteristics using proprietary algorithms that generate composite BC risk scores designed to assist in BC diagnosis and disease characterization, predicting response to BCG therapy, and tracking treatment progress. The Oncuria assay is currently available in the USA [16,17]. In 2024, we published a proof-of-concept study that demonstrated similar Oncuria performance across three different Luminex instrumentation platforms commonly used by diagnostic clinical laboratories [18]. However, that report was constrained by a limited cohort size comprising urine specimens from 36 subjects (18 with BC). The current study, following regulatory guidance [19], compared Oncuria assay performance across the same three analyzers in an expanded cohort comprising 181 subjects (54 with BC).

## 2. Materials and Methods

### 2.1. Subjects and Urine Samples

Subjects included 54 individuals with BC and 127 non-BC controls. Data are reported according to the PROBE criteria [20]. There was no overlap between the current cohort and previously published cohorts [14]. The primary exclusion criterion was renal insufficiency (i.e., GFR <60 mL/min), because this condition is associated with proteinuria that can interfere with protein immunoassay performance and interpretation. Midstream voided urine samples that had been collected for cytology were centrifuged at 1000× *g* for 10 min to sediment particulates, and supernatants were immediately frozen to −20 °C—only a single freeze–thaw cycle occurred before the Oncuria evaluation.

This study received approval and a waiver of consent to use previously banked anonymized urine samples from the Cedars-Sinai Medical Center Institutional Review Board, Los Angeles, CA, USA (IRB #00001459). All samples were originally collected under IRB-approved protocols with written informed consent from participants for specimen banking and future research use. Study performance complied with the tenets of the Declaration of Helsinki.

### 2.2. Oncuria Assay

The multiplex Oncuria bead-based fluorescence assay simultaneously evaluates 10 protein analytes [serpin A1/alpha 1 anti-trypsin (A1AT), angiogenin (ANG), apolipoprotein E (ApoE), carbonic anhydrase IX (CA9), CXCL8/interleukin-8 (IL-8), matrix metalloproteinases-9 (MMP-9) and −10 (MMP-10), serpin E1/plasminogen activator inhibitor-1 (PAI-1), CD138/syndecan-1 (SDC-1), and vascular endothelial growth factor-A (VEGF-A)] in urine samples, using Luminex xMAP^®^ (Multiple Analyte Profiling) technology (Luminex Corp.). With Oncuria, 10 distinct capture bead sets (panel #1 MMP-9, IL-8, VEGF-A, CA9; panel #2 A1AT, ANG, APOE, PAI-1, SDC-1; panel #3 MMP-10) allows the 10 target analytes to be concurrently isolated and analyzed by incubation with a single urine sample. Each bead set contains uniquely color-coded (thus differentiable) 6.5 µm polystyrene beads that are coupled to antibodies against a specific target biomarker. The beads’ paramagnetic core allows easy bead retrieval using a magnet. Beads with their captured target antigens are then quantified by flow cytometric analysis (Luminex 200 and FlexMap 3D; Luminex, Austin, TX, USA) and LED image analysis (MagPix; Luminex, Austin, TX, USA). Oncuria is in clinical trials to obtain FDA clearance/approval as an in vitro diagnostic test for detecting new BC cases in patients with hematuria (Oncuria-Detect) [21,22], for predicting responsiveness to BCG treatment in patients with BC (Oncuria-Predict) [23], and for identifying recurrence in patients with history of BC (Oncuria-Monitor) [24]. In a 2021 clinical validation study to detect de novo BC, the assay demonstrated an area under receiver operating curve, AUROC, value of 0.95 (95% CI: 0.90–1.00), with 93% specificity and 93% sensitivity, and an NPV of 0.99 and a PPV of 0.65 [12]. In a 2022 report, Oncuria predicted patient responsiveness to intravesical BCG therapy for treating NMIBC with an AUROC of 0.89, a sensitivity of 82%, and a specificity of 85% [13]. In a 2024 proof-of-concept report of 36 subjects (18 with BC), Oncuria urinary analyte concentrations were essentially identical when measured across three common analyzers [18]. The current report compared Oncuria performance across three analyzer instrumentation systems using a much larger cohort.

### 2.3. xMAP Instrumentation

Oncuria assay performance was compared using the laser/flow-based Luminex 200 and FlexMap 3D xMAP instruments and the LED/image-based MagPix. Analyzer software was the xPONENT software V4.3 (Luminex 200) and V4.2 (MagPix and FlexMap 3D) from Luminex Corp [25]. The Model 200 analyzer accommodates multiplex analysis of up to 100 analytes per sample and reads 96-well microtiter plates in ≤1 h. The more portable MagPix instrument can simultaneously evaluate 50 analytes in 96-well plates in ≈1 h. Both the 200 and MagPix models provide ≥ 3.5 logs of dynamic range. The newer FlexMap 3D system can measure up to 500 analytes per sample. It has an increased dynamic range (≥4.5 logs) versus earlier analyzers, and uses automated high-throughput analysis to read 96-well plates in ≈20 min and 384-well plates in ≈75 min [25].

### 2.4. Sample Treatment

Voided urine samples were thawed at 4 °C and centrifuged at 15,000× *g* for 5 min to sediment any particulates. Samples, standards, and controls (50 μL/well) were assayed in duplicate wells in 96-well plates. Standard curves for all 10 analytes were 3-fold dilutions that covered the dynamic range (>3-log) of all analytes. Bead incubations with sample and standards were performed in 96-well plates; at room temperature for 2 h, beads with captured analytes were immobilized, washed, resuspended, and analyzed.

### 2.5. Data Analysis

Data were analyzed using the SAS statistical software version 9.4 (SAS Institute Inc., Cary, NC, USA). Analyte concentrations were determined using standard curves generated with a 5-parameter logistical curve fit algorithm available with the xPONENT software. Biomarker concentrations are shown as mean pg/mL ± SD, median values, range, or n (%) of samples, as applicable. The intraclass correlation for inter-rater reliability assessed the agreement among the three instruments. For protein calculations, sample analyte measurements below the lowest standard were entered as the lowest standard concentration for that analyte, and values above the highest standard were replaced with that analyte’s respective highest standard value.

## 3. Results

### 3.1. Analyte Detection Ranges

The dynamic range of quantification (the lowest to highest standard concentration) for the 10 analytes are shown in Table 1 according to Certificate of Analysis enclosed in the kit. The lowest quantification limit of the 10 concurrently evaluated analytes was for CA9, at 1.5 pg/mL. The highest upper quantification limit was for A1AT, at 445,620 pg/mL.

### 3.2. Subject Characteristics

Subject demographics are shown in Table 2. Most subjects were ≥65 years (58%) and male (70%). Self-identified race was approximately evenly divided between White and non-White. Of subjects with BC, 76% (n = 41/54) had NMIBC, and disease was classified as high-grade in 76% (n = 41/54).

### 3.3. Signal Strength by Instrument

Raw fluorescence signals from the Model 200 and MagPix systems were nearly identical for all 10 analytes, and were both lower than raw signals from the FlexMap 3D analyzer (Appendix A), in agreement with previously published observations [18]. This variation stems from variations in the optical analysis platforms used by the different systems, and has no impact on calibration curve characteristics or analyte measurements.

### 3.4. Biomarker Quantification by Instrument

The calculated average concentrations of all 10 analytes were very close to identical, across all three instruments, in 100% (181/181) of urine samples (Table 3, Figure 1), with all intraclass correlations >0.99. A1AT had urinary concentrations that exceeded the upper standard of its dynamic range, observed with all three analyzers in 6% (n = 11/181) of patient samples, i.e., 6/54 cancer subjects and 5/127 controls. Interleukin-8 levels exceeded the highest standard value in 11% (6/54) of cancer subjects and 2% (2/127) of controls. ANG exceeded the highest standard curve value in 11% (6/54) of cancer subjects and 0% of controls. MMP-9 exceeded the highest standard value in 2% (2/54) of cancer subjects and 0% of controls. We acknowledge high SD values for all analytes—this was due to the wide range of measured concentrations across our study cohort that pooled healthy subjects with cancer patients.

While this report is intended to demonstrate assay reproducibility across different xMAP instruments and not for clinical validation, the noteworthy elevations of several analytes were noted in the representative urine samples for cancer versus control (Table 4 and Appendix A). Clinical studies that validated the Oncuria assay for diagnosing BC have been published elsewhere [12]. In the current cohort, all 10 urinary analyte levels trended higher in cancer versus control subjects.

### 3.5. Correlation Between Concentration Measurements from Different Analyzers

Scatter plots for all analytes were generated to evaluate the correlation between concentration measurements from three instruments (Appendix A). When comparing the three instruments’ output pairwise, excellent correlation existed between calibration curves for all analytes, with R^2^ (coefficient of determination) values of 0.999 for 15 of 30 comparisons and R^2^ ≥ 0.995 for 27 of 30 comparisons, while the range of concentrations were wide (Table 5). The largest variation occurred with CA9, with R^2^ values ranging from 0.948 to 0.970, and associated %CVs ranging from 17.4% to 23.1%; most urinary CA9 concentrations were near the low end of the dynamic range, with multiple outliers at high concentration.

## 4. Discussion

The multiplex Oncuria BC assay performed equivalently across three different Luminex flow-based analyzers, for simultaneously evaluating its 10 biomarkers in urine samples to develop cancer risk signatures. Such agreement across different xMAP instrumentation platforms that are in common laboratory use supports the feasibility of performing the Oncuria assay in diverse diagnostic laboratories using popular equipment. The Oncuria assay is a noninvasive and rapid-reporting supplement to cystoscopy for identifying BC, predicting disease response to medical therapies, tracking treatment progress, and monitoring for neoplastic recurrence [12,13,14,15]. The Oncuria assay was previously demonstrated to perform equivalently well on these three analyzers [18], but that study was restricted to 36 participants, of whom only 18 had a BC diagnosis. The current study greatly strengthens those initial finding in an expanded cohort of 181 subjects, including 54 individuals with diagnosed BC.

Using multiplex technology to generate complex urinary protein profiles has benefits over single-protein assessment for detecting BC [9]. All three of the current FDA-approved single-plex urinary protein BC assays lack the 10-plex Oncuria assay’s overall 93% sensitivity for identifying BC, including an 89% sensitivity for capturing low-grade BC [12]. In comparison, the commercial nuclear matrix protein-22 (NMP22 BladderChek^®^) [10], bladder tumor antigen (BTA-Stat^®^ and BTA-Trak^®^) [11], and mini chromosome maintenance-5 (MCM5) protein (ADxBladder^®^) [26] single-plex assays have lower respective sensitivities of 25%, 36%, and 50% for detecting low-grade BC. None of these three markers are specifically expressed by BC cells, and test interpretation may be confounded by the fact that both urinary BTA and NMP22 levels can be elevated during urinary tract inflammation unrelated to BC [10,11]. The Oncuria multiplex protein profiling assay can enhance diagnostic accuracy by generating a complex BC urinary analyte signature that better characterizes disease risk than individual biomarkers. The Oncuria assay simultaneously interrogates urine samples for levels of 10 BC-related analytes, and its validated algorithm incorporates patient characteristics (i.e., age, sex, race) to rapidly generate an accurate BC risk stratification score [18]. The Oncuria algorithm is undergoing continuous refinement by considering new patient parameters and medical historical details to additionally enhance its value in providing personalized cancer care [27]. Multiplex testing can reduce labor requirement compared to measuring several biomarker levels individually [28]. The high-throughput capacity of contemporary automated xMAP instrumentation platforms improves laboratory efficiency and speeds up sample-to-results timing [25].

Cystoscopy and voided urine cytology persist as gold-standard approaches for BC clinical evaluation [3,4]. Cystoscopy is an invasive procedure with an attendant risk of adverse events, and these risks are increased in older patients. Also, while white-light cystoscopy can detect papillary lesions with high sensitivity, it may overlook flat lesions (e.g., carcinoma in situ, CIS); however, alternative imaging techniques (e.g., fluorescence cystoscopy, narrow-band imaging, etc.) can better distinguish tumor tissue including CIS [29]. There is no universally accepted protocol for endoscopic follow-up of patients with NMIBC bladder cancer or those with MIBC who underwent bladder-sparing radiotherapy [30]. Cystoscopy may be overused in up to 75% of patients with low-risk NMIBC [31], and high-intensity cystoscopic surveillance does not improve 5-year outcomes in high-risk NMIBC [32]. Overuse of cystoscopy increases both healthcare costs and risk of treatment-related adverse events. Frequent cystoscopy among patients with low-risk NMIBC is associated with twice as many transurethral resections and does not decrease the risk for bladder cancer progression or death [33]. A major benefit of the noninvasive Oncuria assay’s high NPV (99%) [12] is that a determination of low-risk of BC may reduce the need for frequent invasive cystoscopic procedures.

Voided urine cytology and Oncuria are both noninvasive diagnostic approaches. Although voided urine cytology is also economical, it tends to miss early and low-grade BC [5]. However, the implementation of the Paris System for Reporting Urinary Cytology, created in 2016, has improved cytology’s performance in identifying high-grade BC by reducing the rate of indeterminate (atypical) assessments [5]. Cytology has a reported overall sensitivity of 48% for identifying BC (84% for detecting high-grade tumors versus only 16% for low-grade tumors) [34]. The Oncuria assay previously demonstrated 94% sensitivity for detecting high-grade BC and a notable 89% sensitivity for identifying low-grade BC [12]. Cytology alone has a reported overall specificity of 86% for identifying BC [34], whereas Oncuria previously demonstrated 93% specificity [12].

A major study strength is the enhanced generalizability of findings by evaluating an expanded cohort comprising adults of all ages, including a substantial proportion of females, of heterogeneous racial background. This cohort diversity has clinical relevance because the Oncuria algorithm incorporates these specific patient characteristics (i.e., age, race, and gender) into the 10-biomarker analysis to calculate BC risk scores [12]. The main study limitation was that analytical comparisons were limited to three contemporary xMAP instrument systems manufactured by a single company. Besides the fact that Luminex instrumentation has good penetration within diagnostic laboratories worldwide, bead-based multiplex immunoassays can be analyzed using other fluorescence detection instruments, which may facilitate homogeneous assay performance across different facilities [34]. Another study limitation is that this analysis was not designed as a clinical validation study, but rather to ensure technical reproducibility across platforms, which is critical for clinical assay deployment. Oncuria clinical validation studies have been previously published [12,14,15].

## 5. Conclusions

The multiplex Oncuria BC assay performed similarly well across three different contemporary analyzers for determining 10 biomarker BC signatures in voided urine specimens. This concordant performance across fluorescence detection platforms suggests that the Oncuria assay can be easily integrated into diagnostic laboratories that employ xMAP technology, and may be compatible with other fluorescence analyzers. The Oncuria assay offers a fast and noninvasive diagnostic approach that has clinical value by supplementing cystoscopy for accurately identifying BC, predicting BCG treatment response, and surveilling for treatment effectiveness and disease recurrence.

## Figures and Tables

**Figure 1 diagnostics-15-01749-f001:**
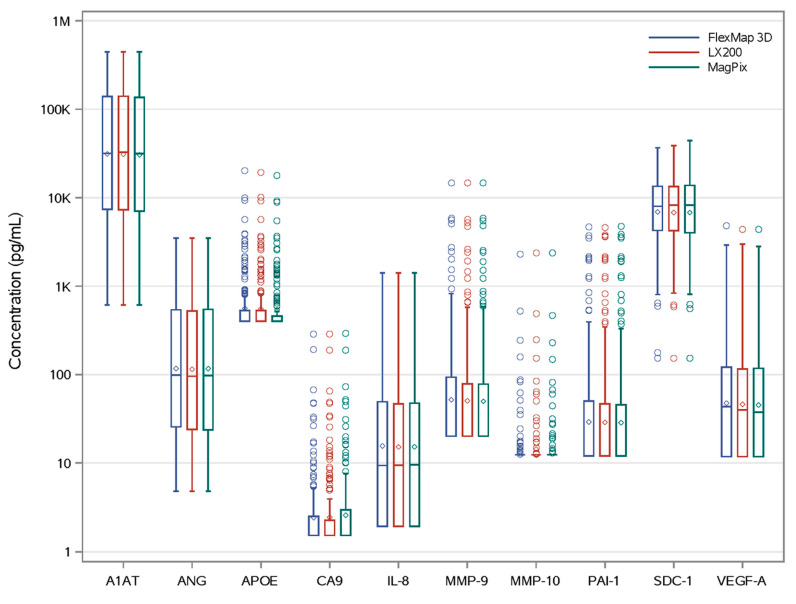
Oncuria analyte concentrations across three analyzers. All 10 urinary analytes were quantified nearly identically on the three Luminex xMAP analyzers. Box and whiskers indicate median analyte concentrations (pg/mL; horizontal lines within boxes) and interquartile ranges. Diamond symbols (“◇”) indicate mean concentrations, and round symbols (“○”) indicate outliers (concentration >1.5× interquartile range boundaries Q3–Q1).

**Table 1 diagnostics-15-01749-t001:** The dynamic range of the 10 bioanalytes in the Oncuria assay.

Biomarker	Lowest Standard, pg/mL	Upper Standard, pg/mL
A1AT	611	445,620
ANG	4.8	3500
APOE	400	291,720
CA9	1.5	1110
IL-8	1.9	1410
MMP-9	20.1	14,680
MMP-10	12.3	9030
PAI-1	12.0	8760
SDC-1	153	111,500
VEGF	11.8	8620

**Table 2 diagnostics-15-01749-t002:** Subject characteristics.

Parameter	All (N = 181)	Controls, N = 127	Bladder Cancer, N = 54	*p*-Value
Age, years, mean (range)	65.5 (26–88)	64.2 (26–89)	68.9 (52–88)	0.02
18–54 years, n (%)	31 (17.1)	27 (21.3)	4 (7.4)	
55–64 years, n (%)	46 (25.4)	33 (26.0)	13 (24.1)	
65–74 years, n (%)	61 (33.7)	38 (29.9)	23 (42.6)	
≥75 years, n (%)	43 (23.8)	29 (22.8)	14 (25.9)	
Male/female ratio (% male)	126:55 (69.6% male)	85:42 (66.9% male)	41:13 (75.9% male)	0.23
Race, n (%)				0.41
White	92 (50.8)	68 (53.5)	24 (44.4)	
Other	88 (48.6)	58 (45.7)	30 (55.6)	
Unknown	1 (0.6)	1 (0.8)	0 (0.0)	
Primary Tumor Stage, n (%)				N/A
NMIBC (Ta, Tis, or T1)		N/A *	41 (76%)	
MIBC (T2–T4)		N/A	13 (24%)	
Grade, n (%)				N/A
Low		N/A	13 (24%)	
High		N/A	41 (76%)	

* N/A, not applicable.

**Table 3 diagnostics-15-01749-t003:** Similar mean analyte measurements across analyzers.

Analyte (pg/mL)	FlexMap 3D (N = 181)			LX200 (N = 181)			MagPix (N = 181)			
	Mean	SD	Median	Mean	SD	Median	Mean	SD	Median	ICC *
A1AT	95,335	1.3 × 10^5^	31,412	98,525	1.4 × 10^5^	31,172	98,052	1.3 × 10^5^	31,168	>0.99
ANG	472	784	99	468	782	96	473	781	98	>0.99
APOE	888	1874	400	866	1813	400	829	1676	400	>0.99
CA9	7	26	2	7	26	2	7	27	2	>0.99
IL-8	155	350	8	163	359	9	158	292	10	>0.99
MMP9	397	1701	20	387	1686	20	388	1692	20	>0.99
MMP10	32	176	12	32	181	12	31	174	12	>0.99
PAI-1	198	656	12	198	658	12	196	658	12	>0.99
SDC-1	9540	6848	8005	9470	6934	8209	9553	7208	8241	>0.99
VEGF	165	499	43	160	476	40	159	466	38	>0.99

Values rounded to the nearest whole value. Measurements below/above the lowest/highest standard curve values were assigned the respective standard value. * ICC is the intraclass correlation for inter-rater reliability.

**Table 4 diagnostics-15-01749-t004:** Biomarker protein concentrations in urine samples compared across three flow analyzers (pg/mL).

Sample ID	Instrument	A1AT	ANG	ApoE	CA9	IL-8	MMP-9	MMP-10	PAI-1	SDC-1	VEGF
#M003	FlexMap 3D	340,684	3500	9321	48	1410	934	519	4642	31,386	4836
Cancer	200	324,637	3500	9104	47	1410	863	491	4599	31,382	4397
	MagPix	328,482	3500	8861	52	1410	871	462	4756	33,713	4353
#M004	FlexMap 3D	421,933	464	851	12	1410	300	12	271	26,923	513
Cancer	200	348,662	441	830	13	1410	280	12	268	28,563	497
	MagPix	409,455	526	769	20	1410	279	12	280	32,092	526
#M006	FlexMap 3D	170,574	835	922	3	463	49	12	284	36,675	180
Cancer	200	178,759	874	886	6	444	46	12	292	38,813	174
	MagPix	256,081	1001	845	3	468	41	12	303	44,261	182
#C017	FlexMap 3D	5528	98	400	2	2	20	12	12	3467	12
Control	200	5544	95	400	2	2	20	12	12	3463	12
	MagPix	5286	96	400	2	2	20	12	12	3353	12
#C018	FlexMap 3D	3557	6	400	2	2	20	12	12	1102	12
Control	200	3526	8	400	2	2	20	12	14	1039	12
	MagPix	3100	7	400	2	2	20	12	12	995	12
#C019	FlexMap 3D	69,062	174	400	2	4	20	12	12	3698	47
Control	200	71,593	166	400	2	3	20	12	12	3525	48
	MagPix	71,237	166	400	2	3	20	12	12	3573	46

**Table 5 diagnostics-15-01749-t005:** Summary of scatter plots comparing data output between analyzers.

Biomarker	Instruments	n	RMSE	R^2^	%CV
A1AT	LX200 vs. MagPix	181	1.05	0.999	0.5%
	LX200 vs. FlexMap 3D	181	1.05	0.999	0.4%
	MagPix vs. FlexMap 3D	181	1.07	0.999	0.6%
ANG	LX200 vs. MagPix	181	1.07	0.999	1.4%
	LX200 vs. FlexMap 3D	181	1.08	0.998	1.7%
	MagPix vs. FlexMap 3D	181	1.08	0.998	1.6%
ApoE	LX200 vs. MagPix	181	1.03	0.997	0.5%
	LX200 vs. FlexMap 3D	181	1.02	0.999	0.4%
	MagPix vs. FlexMap 3D	181	1.04	0.997	0.6%
CA9	LX200 vs. MagPix	181	1.17	0.970	17.4%
	LX200 vs. FlexMap 3D	181	1.20	0.960	20.4%
	MagPix vs. FlexMap 3D	181	1.24	0.948	23.1%
IL-8	LX200 vs. MagPix	181	1.04	0.999	1.6%
	LX200 vs. FlexMap 3D	181	1.06	0.999	2.2%
	MagPix vs. FlexMap 3D	181	1.07	0.999	2.3%
MMP-9	LX200 vs. MagPix	181	1.03	0.999	0.7%
	LX200 vs. FlexMap 3D	181	1.03	0.999	0.7%
	MagPix vs. FlexMap 3D	181	1.05	0.999	1.2%
MMP-10	LX200 vs. MagPix	181	1.03	0.997	1.2%
	LX200 vs. FlexMap 3D	181	1.03	0.997	1.3%
	MagPix vs. FlexMap 3D	181	1.04	0.995	1.6%
PAI-1	LX200 vs. MagPix	181	1.07	0.998	2.1%
	LX200 vs. FlexMap 3D	181	1.06	0.999	1.6%
	MagPix vs. FlexMap 3D	181	1.07	0.998	2.0%
SDC-1	LX200 vs. MagPix	181	1.03	0.999	0.3%
	LX200 vs. FlexMap 3D	181	1.02	0.999	0.3%
	MagPix vs. FlexMap 3D	181	1.04	0.999	0.4%
VEGF	LX200 vs. MagPix	181	1.06	0.998	1.6%
	LX200 vs. FlexMap 3D	181	1.06	0.998	1.5%
	MagPix vs. FlexMap 3D	181	1.07	0.998	1.7%

RMSE = root mean square error; %CV = percent coefficient of variation.

## Data Availability

The anonymized datasets used and/or analyzed during the current study are available from the corresponding author upon reasonable request.

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
