# Peer review of "Expanded Performance Comparison of the Oncuria 10-Plex Bladder Cancer Urine Assay Using Three Different Luminex xMAP Instruments"

_diagnostics, 2025, doi:10.3390/diagnostics15141749_

Round 1

Reviewer 1 Report

Comments and Suggestions for Authors
  1. How do the authors avoid the protease activity in the urine sample? At what temperature are the collected urine samples frozen?
  2. The website does not exist in reference 19.
  3. In Table 1, which instrument is used?
  4. In Table 3, the standard deviation (SD) is high. Why do the authors use samples of cancer urine mixed with a control?
  5. According to Table 4, are the xMAP analyzers suitable for quantifying CA9 and MMP-10? The xMAP analyzer's quantification may cause false negatives.

Reviewer 2 Report

Comments and Suggestions for Authors

Tanaka et al. present a comparison of the Oncuria 10-plex bladder cancer urine assay using 3 Luminex xMAP instrument models (FlexMap 3D, FlexMap 200, MagPix) and urine samples from a total of 181 subjects, 51 of whom were bladder cancer patients. They observed excellent agreement between results obtained for the 3 instruments. They discuss the utility of the assay as a noninvasive tool to complement cytology and cystology for bladder cancer diagnosis, prognosis and follow-up.

The manuscript is totally focused on a technical question that was addressed in a prior study that employed 36 samples.  Nevertheless it is well-written and the results are clearly presented.

Major comment

Figure 1 compares analyte concentrations measured with the 3 instruments for all of the urine samples grouped together. To make the manuscript more interesting to a broader audience of researchers/clinicians, the Authors should add a graph that compares urine analytes in the bladder cancer patients and controls, either as a second main figure or as a supplementary figure.

Minor comment

On line 39, the abbreviation ‘CLIA’ (Clinical Laboratory Improvement Amendments) should be defined.

Reviewer 3 Report

Comments and Suggestions for Authors

Dear Auhors,

I read with interest your article, however, I found some points that need to be expanded.

1) You proved that Oncuria BC urine assay have reliable outcomes. However, as a clinician, I would like to have also the sensitivity and specificity of the samples collected in this study. So in how many patients the assay correctly proved that the sample was BCa or healthy. Please add it in the results

2) Strength and limitations of this study should be implemented also with a mention of the clinical results of the assay 

3) Please make a mention of the IRB and Ethics appraisal also in the article body
